# Novel Protein-Based Vaccine against Self-Antigen Reduces the Formation of Sporadic Colon Adenomas in Mice

**DOI:** 10.3390/cancers13040845

**Published:** 2021-02-17

**Authors:** Elodie Belnoue, Alyssa A. Leystra, Susanna Carboni, Harry S. Cooper, Rodrigo T. Macedo, Kristen N. Harvey, Kimberly B. Colby, Kerry S. Campbell, Lisa A. Vanderveer, Margie L. Clapper, Madiha Derouazi

**Affiliations:** 1AMAL Therapeutics, Fondation pour Recherches Médicales, 64 avenue de la Roseraie, 1205 Geneva, Switzerland; elodie.belnoue@boehringer-ingelheim.com (E.B.); susanna.carboni@boehringer-ingelheim.com (S.C.); 2Boehringer Ingelheim International GmbH, 55216 Ingelheim, Germany; 3Cancer Prevention and Control Program, Fox Chase Cancer Center, 333 Cottman Ave, Philadelphia, PA 19111, USA; alyssa.leystra@fccc.edu (A.A.L.); harry.cooper@fccc.edu (H.S.C.); rodrigo.macedo@fccc.edu (R.T.M.); Kristen.harvey@fccc.edu (K.N.H.); lisa.vanderveer@fccc.edu (L.A.V.); 4Department of Pathology, Fox Chase Cancer Center, 333 Cottman Ave, Philadelphia, PA 19111, USA; 5Blood Cell Development and Function Program, Fox Chase Cancer Center, 333 Cottman Ave, Philadelphia, PA 19111, USA; kcolby@its.jnj.com (K.B.C.); kerry.campbell@fccc.edu (K.S.C.)

**Keywords:** cancer vaccine, colon cancer, mouse model, Ascl2, T cell response, humoral response

## Abstract

**Simple Summary:**

Colorectal cancer remains a leading cause of cancer-related mortality worldwide. However, high-risk populations with a genetic predisposition for colorectal cancer could benefit greatly from novel and efficacious immunopreventive strategies that afford long-lasting protection. The achaete-scute family bHLH transcription factor 2 (Ascl2) has been identified as a promising target for immunoprevention of colorectal cancer, based on its induction during the formation and progression of colorectal tumors and its minimal expression observed in healthy tissue. The goal of the present study was to determine the efficacy of a protein-based vaccine targeting Ascl2 in combination with an anti-PD-1 treatment in a spontaneous colorectal cancer mouse model. This novel vaccine strategy promotes potent tumor-specific immunity, and prevents the formation of colon adenomas in mice. The results demonstrate that Ascl2 is a promising target for immunoprevention for individuals at elevated risk of developing colorectal cancer.

**Abstract:**

Novel immunopreventive strategies are emerging that show great promise for conferring long-term protection to individuals at high risk of developing colorectal cancer. The KISIMA vaccine platform utilizes a chimeric protein comprising: (1) a selected tumor antigen; (2) a cell-penetrating peptide to improve antigen delivery and epitope presentation, and (3) a TLR2/4 agonist to serve as a self-adjuvant. This study examines the ability of a KISIMA vaccine against achaete-scute family bHLH transcription factor 2 (Ascl2), an early colon cancer antigen, to reduce colon tumor formation by stimulating an anti-tumor immune response. Vaccine administrations were well-tolerated and led to circulating antibodies and antigen-specific T cells in a mouse model of colorectal cancer. To assess preventive efficacy, the vaccine was administered to mice either alone or in combination with the immune checkpoint inhibitor anti-PD-1. When delivered to animals prior to colon tumor formation, the combination strategy significantly reduced the development of colon microadenomas and adenomas, as compared to vehicle-treated controls. This response was accompanied by an increase in the intraepithelial density of CD3+ T lymphocytes. Together, these data indicate that the KISIMA-Ascl2 vaccine shows great potential to be a safe and potent immunopreventive intervention for individuals at high risk of developing colorectal cancer.

## 1. Introduction

Colorectal cancer (CRC) remains a leading cause of cancer-related mortality in the US and Europe [1]. Although the benefits of early detection are well documented, only 63% of the US population adhere to current screening guidelines [2,3]. Even among those who do comply, as many as 25% of colon adenomas are missed during routine endoscopic surveillance. These disturbing statistics, when combined with a recent rise in the incidence of CRC among young adults [4], dictate the need for new intervention strategies to reduce the morbidity and mortality associated with this disease.

Several high-risk populations with a genetic predisposition for CRC have been identified who would benefit greatly from an early intervention that affords durable, long-lasting protection. The lifetime risk of developing CRC is 50–80% for individuals who inherit a germline mutation in the mismatch repair genes (Lynch Syndrome) and 100% for individuals born with a mutation in the APC gene (Familial Adenomatous Polyposis (FAP)) [5]. Even individuals with a first-degree relative with CRC face a 2–3 fold increased risk of a similar diagnosis, as compared to the general population [5]. The development of efficacious preventive interventions for these high-risk populations has been hindered by the discovery that many of the effective chemopreventive agents also cause serious side effects [6,7]. However, immunoprevention shows great promise for high-risk populations, including those at increased risk for CRC [8,9]. Results from a phase I/II trial of an immunopreventive vaccine against MUC1 (NCT-007773097) indicate immunogenicity in 44% of vaccinated individuals and no significant toxicity [10]. Although these exciting results underscore the potential of cancer preventive vaccines, alternative targets may provide greater immunogenicity and/or protection. 

The achaete-scute family bHLH transcription factor 2 (Ascl2) is a promising target for immunoprevention of CRC for several reasons. First, Ascl2 participates in Wnt signaling, a pathway activated in the majority (>80%) of sporadic and familial colorectal cancers [11]. As a WNT-responsive transcription factor, Ascl2 acts as a master regulator of intestinal stem cell identity [12]. Second, Ascl2 is induced during the formation and progression of colorectal tumors, with minimal expression observed in non-neoplastic tissue [13,14,15,16]. Third, preventive vaccination of *Apc^+/Min-FCCC^* mice with a recombinant Ascl2 peptide plus AS15 adjuvant caused a 3–4 fold reduction in colorectal microadenomas in the absence of a CD8 T cell response [16]. Although promising, these data suggest that additional protection might be gained from an alternative vaccine strategy. 

An optimal cancer vaccine should simultaneously stimulate cytotoxic T cell-mediated immunity, induce T helper cells, prime a multi-antigenic immune response for different HLA restriction and promote immunological memory. A novel cancer vaccine platform, named KISIMA^TM^ [17], addresses the above listed requirements for a robust and prolonged immune response. KISIMA^TM^ vaccine is a recombinant chimeric protein composed of three elements. First, a cell-penetrating peptide (CPP) enhances vaccine delivery and allows epitope presentation on both major histocompatibility complex (MHC) class I and II molecules [18,19]. Second, an antigenic domain is rationally designed with selected antigen portions containing MHC class I and class II epitopes from different tumor-specific antigens across a range of HLA restrictions. A TLR2/4 agonist is the third component, conferring self-adjuvant activity to the vaccine. 

The KISIMA^TM^ platform allows CPP-mediated antigen delivery to the antigen presenting cells (mainly dendritic cells), while simultaneously activating the given dendritic cell via a TLR peptide agonist. This results in the activation of potent CD8 and CD4 T cell immune responses in preclinical models [17]. The consequent high infiltration of cytotoxic CD8 T cells within the tumor mass promotes the efficacious control of tumor development in different mouse models [17].

The *Apc^+/Min-FCCC^* mouse model is ideal for immunoprevention studies, in particular testing novel vaccines against Ascl2 [16,20]. This immunocompetent animal carries a heterozygous mutation in the Apc gene that renders it susceptible to spontaneous colon tumorigenesis. Tumors form the following molecular and histological processes similar to those observed in humans; loss of the second copy of Apc in colonic epithelial cells results in increased Wnt signaling, activation of β-catenin-mediated TCF signaling, and the formation of microadenomas and multiple colonic adenomas. As in humans, colorectal adenomas in the *Apc^+/Min-FCCC^* model overexpress the murine form of Ascl2, murine achaete scute homolog 2 (Mash2) [16].

The goal of the present study was to determine the efficacy of a novel Ascl2 targeted KISIMA vaccine in stimulating an immune response and inhibiting the formation of colon adenomas in *Apc^+/Min-FCCC^* mice.

## 2. Materials and Methods

### 2.1. Animals

Female C57BL/6J mice were purchased from Charles River Laboratories (L’Arbresles, France) and used between 6 and 12 weeks of age at the time of experiments at AMAL Therapeutics. Mouse studies were reviewed and approved by the institutional and cantonal veterinary authorities in accordance with Swiss Federal law on animal protection.

Male C57BL/6 *Apc^+/Min-FCCC^* mice were obtained from a colony at Fox Chase Cancer Center (FCCC) that had been maintained for at least 66 generations prior to initiation of the study [20]. These animals carry a heterozygous nonsense mutation in codon 850 of the *Apc* tumor suppressor gene. All experiments were approved by the Institutional Animal Care and Use Committee at FCCC on 24 January 2018. Additional information about animal housing and maintenance is available in the Appendix A.

### 2.2. Colonoscopy

A total of 126 *Apc^+/Min-FCCC^* mice were enrolled on study 4–5 days prior to receiving the first injection at 40 ± 5 days (mean ± standard deviation; range 31–49 days) of age. Animals weighing >15 g (*n* = 69) were examined by colonoscopy. Following bowel preparation, colonoscopic examinations were performed using a veterinary endoscope (1.5 mm outer diameter) (Karl Storz Veterinary) with a 0° viewing angle as described previously [21]. Baseline tumor status was recorded for each animal. All animals undergoing colonoscopy were categorized as ‘tumor-free’ (no tumors observed) (*n* = 64), or ‘tumor-bearing’ (at least one tumor identified) (*n* = 5). Animals weighing ≤ 15 g (*n* = 57) were not subjected to colonoscopy and categorized as ‘not determined’.

Twenty animals (15 tumor-free, 1 tumor-bearing, and 4 not determined) were enrolled onto the immunogenicity study; the remaining animals (49 tumor-free, 4 tumor-bearing, and 53 not determined) were enrolled onto the efficacy study.

### 2.3. Vaccines

Vaccine construct was designed by AMAL Therapeutics and produced in *E. coli* by Genscript. During the purification process, endotoxins were removed from vaccines through extensive washes with Triton-X114, followed by subsequent affinity chromatography. Endotoxin content was quantified in each vaccine batch using a LAL chromogenic assay. Only the batches with an endotoxin level below 10 EU/mg protein (according to guidelines) were used for further in vivo experiments.

### 2.4. Animal Treatment

Mice were stratified by litter and baseline tumor status, and assigned to one of two treatment arms for the immunogenicity study (Control or Vaccine) or one of four treatment arms (Control, Vaccine, Anti-PD-1, or Vaccine plus Anti-PD-1 (combination)) for the efficacy study using block randomization. Mice included in the analyses (*n* = 17 for the immunogenicity study and *n* = 92 for the efficacy study) were comparable across all treatment arms with respect to age, weight, and tumor status at the time of the first treatment (Appendix A). Biological outliers (defined in “Statistical Analysis” below), including mice that did not complete all scheduled therapies, were excluded from analyses (*n* = 3 for immunogenicity study and *n* = 14 for the efficacy).

Vaccine was administered a total of six times, with 2–4 weeks between injections. Mice in the Vaccine and Combination arms received subcutaneous injections of 100 μL of 20 μM KISIMA-Mash2 vaccine in buffer (50 mM Tris-HCl, 150 mM NaCl, 1 M L-Arginine, 10% Glycerol, pH 8). Mice in the control and anti-PD-1 arms received subcutaneous injections of 100 μL of PBS (immunogenicity study) or vehicle buffer (efficacy study).

In the efficacy study, three cycles of anti-PD-1 were administered over the 17-week treatment schedule. Each cycle consisted of twice-weekly injections. The first cycle was administered 3 days after the second vaccine injection, and was continued for 3.5 weeks. The next two cycles were administered 3 days after the fourth and fifth vaccine injections, and continued for 2.5 weeks each. Mice in the anti-PD-1 and combination arms received intraperitoneal injections of 100 μL of 1 mg/mL anti-Mouse PD-1 (BioXCell #BE0146, lots 665417S1 and 695318A1, Lebanon, NH, USA) in InVivoPure pH 7.0 Dilution Buffer (BioXCell #IP0070, lot 658117D1). Mice in the Control and Vaccine arms received intraperitoneal injections (100 μL) of dilution buffer.

Serum was collected four times throughout the study: prior to the first treatment, one week after the second and fourth vaccine injections, and at the end of the study (Figure 1A). Whole blood was collected by retro-orbital bleed in a non-heparinized capillary blood-collecting tube (Fisherbrand #02-668-15, Waltham, MA, USA). Serum was separated from whole blood in a BD Microtainer SST (Becton Dickinson #365967, Franklin Lakes, NJ, USA), according to the manufacturer’s instructions, and stored at −80 °C until the time of analysis.

Animal health and weight were monitored at least twice per week for the duration of each study. Mice were euthanized by CO_2_ inhalation upon completion of the study, or when exhibiting signs of illness including a hunched body posture, poor grooming, and/or >20% loss of body mass. 

Following euthanasia, spleens were excised, weighed, placed in DMEM containing 100 I.U./mL penicillin and 100 μg/mL streptomycin, and forced through a 70 μM filter. The resulting cell suspension was used for ELISpot analyses in the efficacy study. Organs (liver, kidneys and lungs) were examined for gross alterations. The entire small intestine and colon were excised, opened lengthwise, and washed with PBS. The number of tumors identified upon gross examination, as well as their location and size (length and width as measured with calipers), was recorded. Tumor volume was estimated using an approximation of an ellipsoid ($\mathrm{volume}=\frac{{\mathrm{length}*\mathrm{width}}^{2}}{2}$). The entire colon was placed in 10% buffered formalin overnight for fixation. 

### 2.5. ELISA Assay for Humoral Response

Antibody titers against Mash2 were measured by enzyme-linked immunosorbent assay (ELISA). Plates were coated with Mash2 (5 μg/mL) in PBS, overnight at 4 °C. After washing with PBS-0.05% Tween 20 and blocking with PBS-0.05% Tween 20 containing 1% bovine serum albumin, serial dilutions of sera were added to the plates and incubated at 37 °C for 2 h. After washing, the plates were incubated with peroxidase-conjugated goat anti-mouse IgG (Bethyl Laboratories #A90-131P, Montgomery, TX, USA).

### 2.6. ELISpot Assay

ELISpot assays were performed using the Murine IFN-γ ELISpot Set (Diaclone #862.031 S) according to manufacturer instructions. Splenocytes were isolated with Lympholyte-Mammal (Cedarlane, Burlington, NC, USA) and plated in a concentration gradient from 2.5 × 10^5^–1.0 × 10^6^ cells. Cells were stimulated with vaccine construct or peptide pools, and the assay was completed according to the manufacturer’s instructions. The plate was read using an AID vSPOT Spectrum (Volt Tecnologia #VSR078IFL, Oceanside, CA, USA).

### 2.7. Histopathology

Formalin-fixed colons were cut at ~2 mm intervals, embedded in paraffin, sectioned, and stained with hematoxylin and eosin (H&E). H&E slides were examined by a pathologist who was blinded to treatment group assignment. Tumors were classified as adenomas or microadenomas as described previously [22]. The total number of adenomas included microadenomas.

### 2.8. Immunohistochemistry

Formalin-fixed paraffin-embedded (FFPE) sections (5 µm) were deparaffinized with xylol and hydrated through graded concentrations of alcohol. Sections were then subjected to heat-induced epitope retrieval with 0.01 M citrate buffer (pH 6.0). Endogenous peroxidases were quenched by immersion in 3% hydrogen peroxide. Sections were incubated with rabbit anti-CD3 antibodies (Dako clone A0452, 1:150 dilution) overnight at 4 °C in a humidified slide chamber. Immunodetection was performed using the Dako Envision+ polymer system and visualized with chromogen 3,3’-diaminobenzidine. CD3 positivity (CD3+) was defined as staining highlighting the membrane of a lymphocyte with visible nuclear hematoxylin staining. Overview images of dysplasia and the intervening stroma of each CD3 stained slide were obtained using a Leica LMD6500 microscope. Using QuPath v0.2.2 [23], the dysplastic epithelial cells were outlined together with the intervening stroma and an area (pixel2) calculated. The conversion to mm^2^ was made using the intrinsic LMD6500 scale calibrated with a stage micrometer. Dysplastic burden was defined as the total area (mm^2^) of dysplastic epithelial cells and intervening stroma per animal [24,25]. Intraepithelial CD3+ lymphocytes were defined as CD3+ cells located in-between the dysplastic epithelial cells of adenomas or microadenomas. Lymphocytes shed into the lumen were excluded from evaluation. Stromal CD3+ cells were defined as CD3+ cells located in the stroma immediately adjacent to a dysplastic gland [26]. For each animal, the extent of lymphocyte infiltration was expressed as the number of intraepithelial or stromal lymphocytes, as well as the sum of both elements (total lymphocytes), per lesion or per area (mm^2^) of dysplastic burden.

### 2.9. Statistical Analysis

Biological outliers were defined as mice that did not receive all scheduled therapies (immunogenicity studies *n* = 3; efficacity study *n* = 11) or developed a total number of adenomas > 2 standard deviations from the mean number of adenomas developed in a given treatment group (efficacy study only; 1 Buffer, 1 Vaccine, and 1 Combination-treated). These animals were excluded from all final analyses.

Statistical analyses were performed using Prism software (GraphPad, San Diego, CA, USA) and considered statistically significant if *p* < 0.05.

## 3. Results

### 3.1. KISIMA-Mash2 Vaccine Promotes T Cell and Humoral Responses

A KISIMA vaccine containing the Mash2 protein antigen was designed, and first assessed for its immunogenicity in C57BL/6 mice, which received 6 injections of either vaccine or buffer (control), according to a vaccination schedule established previously [17] and employed in the KISIMA-01 clinical trial (NCT04046445). After six administrations of the KISIMA-Mash2 vaccine, T cell immune response was quantified in the spleen and serum anti-Mash2 IgG titer was determined. A significantly higher T cell immune response against KISIMA-Mash2 vaccine (quantified by IFN-γ ELISpot, Figure 1A) was observed in the spleens of vaccinated mice compared to those of control (buffer-treated) mice. This T cell immune response was associated with a significantly higher antibody titer against Mash2 in vaccinated animals (Figure 1B).

### 3.2. KISIMA-Mash2 Vaccine Is Safe and Promotes an Antibody Response in Apc^+/Min-FCCC^ Mice 

To determine if the KISIMA-Mash2 vaccine is safe and able to promote an immune response in vivo, a small pilot experiment was conducted with male *Apc^+/Min-FCCC^* mice (*n* = 17, 5–8 weeks of age), which received 6 injections of either vaccine (*n* = 9) or PBS (control) (*n* = 8) over the course of 17 weeks (Figure 2A). No difference was observed in the average body weight, appearance, or behavior among mice in all treatment groups (Figure 2B, Appendix A, and data not shown). Strikingly, a strong antibody response against Mash2 was observed in vaccinated *Apc^+/Min-FCCC^* mice (Figure 2C), whereas no antibody response was detected in control *Apc^+/Min-FCCC^* C mice. A trend towards a reduction in the number of gross colon tumors per animal was also observed in vaccinated vs. control animals [27].

### 3.3. Combination of KISIMA-Mash2 Vaccine and Anti-PD-1 Treatment Is Safe and Promotes T Cell and Antibody Response in Apc^+/Min-FCCC^ Mice

Since the KISIMA-Mash2 vaccine promoted some immunogenicity in *Apc^+/Min-FCCC^* mice, a larger study was conducted to determine the ability of the vaccine to prevent colon tumor formation. This study was performed in the presence or absence of the checkpoint inhibitor anti-mouse PD-1 antibody, to potentially enhance the efficacy of the KISIMA-Mash2 vaccine. Male *Apc^+/Min-FCCC^* mice received vaccine alone (*n* = 23), anti-PD-1 alone (*n* = 24), vaccine in combination with anti-PD-1 (*n* = 21), or vehicle (control) (*n* = 27) over the course of 17 weeks (Figure 3A). Again, body weight measurements, as well as the appearance and behavior of the animals, revealed no difference among mice in all treatment groups (Figure 3B, Appendix A, and data not shown), indicating the vaccine was well-tolerated when given alone and in combination with anti-PD-1. *Apc^+/Min-FCCC^* mice vaccinated six times with KISIMA-Mash2 vaccine showed a significant splenic Mash2-specific T cell response, especially with the peptide pool containing the C-terminal portion of Mash2, (i.e., peptide pool 3; Figure 3C), for which the number of predicted class I and class II epitopes was the highest (Appendix A). Interestingly, the T cell immune response was similar in the vaccinated vs. combination groups (Figure 3C), indicating that anti-PD-1 treatment did not improve the immunogenicity of the KISIMA-Mash2 vaccine in the spleen. 

Remarkably, high antibody response against Mash2 was observed in *Apc^+/Min-FCCC^* mice vaccinated with KISIMA-Mash2 alone (Figure 3D) with a peak of antibody response reached after 4 vaccinations. No further benefit was gained from the addition of anti-PD-1. No specific antibody response was detected in *Apc^+/Min-FCCC^* mice treated with vehicle or anti-PD-1 alone.

### 3.4. Combination of KISIMA-Mash2 Vaccine and Anti-PD-1 Treatment Reduces the Formation of Colon Tumors in Apc^+/Min-FCCC^ Mice and Is Associated with Increased T-cell Infiltration into the Intraepithelial Compartment of Colon Adenomas

To determine if the treatment regimen effectively reduced tumor formation, colons from treated vs. control animals were histologically analyzed for the presence of adenomas, including microadenomas. Animals that were tumor-free at the beginning of the study (Appendix A) developed 2.6-fold fewer total colon adenomas following treatment with vaccine and anti-PD-1 combination therapy vs. buffer (*p* = 0.02, Figure 4A). Most striking was a 6.5-fold reduction in the number of microadenomas (*p* = 0.03, Figure 4B) in mice receiving combination therapy vs. buffer. These data indicate that the combination of vaccine and anti-PD-1 reduced the formation of colon tumors, including early precursor lesions, in these animals.

To confirm that the immunogenic effect of the vaccine observed in the spleen of vaccinated mice was associated with an increase in T-cell infiltration into tumors, the number of CD3+ cells in colon adenomas (Figure 5) was examined by immunohistochemistry. The stromal compartment of dysplastic colon tissues exhibited a similar trend for increased CD3+ lymphocyte density in combination-treated vs. control animals (746 ± 110 vs. 733 ± 139 cells/mm^2^ dysplastic tissue; *p* = 0.413, Appendix A). Microadenomas were also examined (Appendix A) and exhibited a trend toward increased numbers of intraepithelial CD3+ cells/lesion in combination-treated (1 ± 0.7 cells/microadenoma) vs. control animals (0.5 ± 0.3 cells/microadenoma; *p* = 0.537). However, the low incidence and small size of these lesions limited the power of the analysis. 

In contrast, within the intraepithelial compartment of colon adenomas, the density of CD3+ lymphocytes was 1.9-fold higher in the group receiving the vaccine and anti-PD-1 combination therapy (179 ± 23 cells/mm^2^ dysplastic tissue), as compared to the control group (96 ± 22 cells/mm^2^ dysplastic tissue) (*p* = 0.028; Figure 5; Figure 6, Appendix A). Interestingly, the density of CD3+ lymphocytes in the combination group was also significantly higher than that of the groups receiving anti-PD-1 alone (*p* < 0.001) or Vaccine alone (*p* < 0.005), demonstrating that single therapy was not sufficient to promote the infiltration of CD3+ cells into the tumor. 

Together, these data indicate that combination therapy with KISIMA-Mash2 vaccine and anti-PD-1 checkpoint blockade increased T-cell infiltration into the intraepithelial compartment of colon adenomas. 

Together, these results demonstrate that KISIMA-Mash2 vaccine, when combined with an immune checkpoint blockade (anti-PD-1 treatment), induces Mash2-specific T cells, which are able to infiltrate the colon and reduce the formation of colon tumors.

## 4. Discussion

The results of the present study indicate that the novel KISIMA-Mash2 vaccine elicits a strong cellular and humoral immune response in mouse models of CRC. When administered in combination with the immune checkpoint inhibitor anti-PD-1, the vaccine decreased the formation of spontaneous colon adenomas in *Apc^+^^/Min-FCCC^* mice free of endoscopically-detectable tumors at baseline. Notably, the combination therapy caused a dramatic (6.5-fold) reduction in the formation of early colon microadenomas. The ability of this vaccine to target putative precursor lesions of CRC indicates the intervention exhibits preventive efficacy very early in the tumorigenesis process.

The present study emphasizes, for the first time, the ability of KISIMA-Mash2 cancer vaccine to induce both antibodies and T cell immune response against a self-antigen, presumably due to the predicted MHC class I- and class II-presented epitopes in Mash2 with potential to stimulate both CD8+ and CD4+ T cells, respectively. In the past, cancer vaccines expected to promote a specific cytotoxic T cell response to fight tumor cells, have often failed. However, new strategies, such as RNA vaccines [28], viral vectors [29,30,31], oncolytic virus [32,33], heterologous prime-boost strategies [34,35,36], or recombinant proteins including the KISIMA vaccine platform [17], have been developed recently. These approaches optimize the induction of potent CD4 and CD8 T cell immune responses, underlining the importance of both components in the fight against cancer. More rarely, cancer vaccines have been proposed to induce antibodies directed against tumor antigen expressed on the surface of cancer cells [37]. Mage-A3 protein vaccine was shown previously to induce clear CD4+ T cell responses that correlate with antibody production but, unfortunately, no convincing evidence of cytotoxic T-cell responses was observed in this trial [38]. In addition, a recent study found that a cytolytic population of CD4+ T cells contributes significantly to anti-tumor immune activity in bladder cancer patients, and their infiltration into metastatic bladder tumors could predict clinical response to anti-PD-L1 therapy [39]. Although a humoral response against the transcription factor Ascl2 was not expected, our results demonstrate the capacity of our vaccine platform to promote a strong humoral response, which is of great interest for developing vaccines against tumor surface antigens, such as Epcam [40] or CEACAM5 [41]. Similarly, the KISIMA vaccine platform may also be a powerful platform for prophylactic vaccines against viral diseases, such as COVID-19 or RSV-related diseases. The strong immune response against the Ascl2/Mash2 self-antigen was associated with strong anti-tumor activity, resulting in a significant reduction in colon microadenomas and adenomas. Ascl2, which is a master regulator of cellular stemness in Lgr5+ cells, is dramatically overexpressed during spontaneous colon tumorigenesis in *Apc^+/Min-FCCC^* mice [16] and in humans (>7-fold in adenomas; >10-fold in carcinomas) [12,13,14]. Expression of Ascl2 is induced by and perpetuates Wnt signaling, a critical event in the formation and progression of most (>80%) CRCs [11,12]. Importantly, no adverse effects were observed when Ascl2 was targeted in this or a previous study despite the important role Ascl2 expression plays in the maintenance of normal Lgr5+ intestinal stem cells [16]. Normal intestinal epithelium may evade immune response due to low expression of Ascl2 (even in intestinal stem cells) as compared to neoplastic intestinal cells [12,15,16,42]. Interestingly, ablation of Lgr5+ stem cells is well-tolerated by healthy mice, resulting in no change in the development of colonic crypts or villi in the small intestine [43,44]. Therefore, even if the vaccine elicits a mild immune response against normal Lgr5+ intestinal stem cells, we would not anticipate a disruption of normal gut homeostasis.

Injection of *Apc^+/Min-FCCC^* mice with a combination of KISIMA-Mash2 vaccine and anti-PD-1 induced the infiltration of CD3+ T lymphocytes into colon adenomas, indicating the treatment regimen successfully mobilized immune cells into the tumor microenvironment. Interestingly, a higher density of tumor-infiltrating lymphocytes (TILs) is associated with improved clinical outcomes in CRC [45]. Furthermore, the density of TILs in colorectal carcinomas is a better predictor of cancer recurrence than the conventional histopathological parameters [46,47]. Therefore, in addition to reducing the formation of colon microadenomas and adenomas, the combination therapy (KISIMA vaccine + anti-PD-1) may slow the rate of progression of any lesions that arise.

Importantly, the addition of anti-PD-1 treatment did not effectively boost the peripheral immune response in vaccinated mice (Figure 3C,D); however, only combination-treated animals exhibited a reduced number of colon microadenomas and adenomas (Figure 4A,B). In contrast, no effect was observed when the mice were only treated with anti-PD1 antibody, which is in agreement with a published report in a very similar model (C57Bl/6J *Apc^Min/+^* mice bearing the same APC mutation), where anti-PD-1 antibody therapy alone had no significant effect on polyp burden [48]. This suggests that the addition of anti-PD-1 can improve vaccine efficacy. This is in line with the emerging notion that cancer vaccines and immune checkpoint inhibitors may function synergistically to induce more effective antitumor immune responses. Although the immune checkpoint inhibitor response rate is high in specific mismatch repair-deficiency (MMR-D)/microsatellite instability-high (MSI-H) phenotypes, this carries unique characteristics of an increased tumor mutational burden and tumor-infiltrating lymphocytes [49]. For the vast majority of colon cancers with low T cell infiltration, use of an immune checkpoint inhibitor in combination with tumor vaccination represents a promising strategy to strikingly enhance T cell infiltration/function and improve clinical response [50,51,52]. 

The safety profile and efficacy of the KISIMA-Mash2 vaccine in the *Apc^+/Min-FCCC^* mouse supports the use of the KISIMA platform, as well as the human Ascl2 antigen as a molecular target, for preventive intervention in populations at elevated risk for CRC. Individuals with a strong family history of CRC are at high risk (2–3-fold) of developing sporadic CRC, and thus represent an ideal population for immunopreventive interventions [5]. As in the *Apc^+/Min-FCCC^* model, most sporadic CRCs arise following the loss of APC, resulting in the formation of microadenomas, adenomas, and ultimately carcinomas with elevated Ascl2 expression [11,13,14,16,20,53]. Thus, this vaccine may provide a way to intervene early in these individuals before the first colon tumors arise.

Although independent analyses are needed, a KISIMA-Ascl2 vaccine may provide protection to individuals with Lynch syndrome who face a >50% lifetime risk of developing CRC and are in urgent need of safe, effective, and durable preventive therapies [5]. We hypothesize the KISIMA-Ascl2 vaccine should be highly effective in these individuals, as (1) progression of most colon polyps in subjects with Lynch Syndrome occurs via the APC pathway [54] as in *Apc^+/Min-FCCC^* mice; (2) early colon lesions in this population exhibit a high mutational burden, which may render these tumors hypersensitive to immunotherapy [22]; and (3) vaccine can be administered to these high-risk individuals prior to tumor development. Additional preclinical studies in mouse models of Lynch syndrome, such as Msh2flox/flox mouse model [55,56], will be required to determine the feasibility of using a vaccine against Ascl2 to prevent colon tumorigenesis in the setting of mismatch repair deficiency. 

While PD-1/PD-L1 blockade has been shown to induce life-saving anti-tumor responses in patients suffering from a wide variety of cancers, its use to enhance vaccination efficacy in healthy individuals is clearly a more provocative concept. Nonetheless, the benefit of using this combination vaccine strategy in those subjects at highest risk for colon malignancy may outweigh the potential risk. Data from this study clearly demonstrate the ability of the combination therapy to provide impressive protection against the formation of sporadic colon tumors in mice. Further study is needed to define the optimal dose and schedule of PD-1/PD-L1 blockade required to elicit a robust cancer vaccine response. The potential use of fewer doses of blocking antibody to confer more effective cancer prevention would further reduce the potential for adverse autoimmune effects in high-risk patient populations, such as individuals with Lynch Syndrome. 

## 5. Conclusions

In conclusion, the results of this study demonstrate that the KISIMA platform elicits a strong and specific immune response, and Ascl2 is a promising target for immunoprevention for individuals at elevated risk of developing CRC.

## Figures and Tables

**Figure 1 cancers-13-00845-f001:**
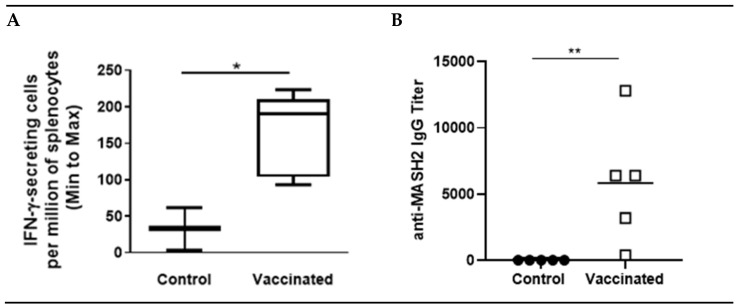
KISIMA-Mash2 vaccine promotes T cell and humoral responses. C57BL/6 mice were treated with 6 administrations of KISIMA-Mash2 vaccine (Day 0, 14, 28, 58, 84 and 127). Thirteen days after the last vaccination, T cell immune response was assessed by ELISpot after stimulation of splenocytes with KISIMA-Mash2 vaccine overnight (**A**). Humoral response was determined in serum by detecting anti-Mash2 IgG by ELISA (**B**). *, *p* < 0.05; **, *p* < 0.01 (unpaired T test).

**Figure 2 cancers-13-00845-f002:**
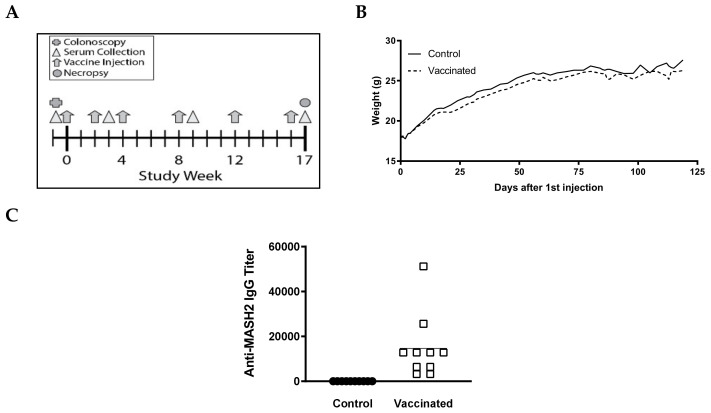
KISIMA-Mash2 vaccine is safe and promotes a humoral response in *Apc^+/Min-FCCC^*mice. (**A**) Study design. *Apc^+/Min-FCCC^* mice underwent colonoscopy (cross) prior to enrollment on study. Serum was collected (triangles) at 3 different time-points. Vaccine or vehicle control (arrows) was administered six times. All animals were necropsied one week after the last vaccine injection. (**B**) Average body weight of animals obtained weekly for the duration of the study. (**C**) Humoral response determined in serum at Week 17 by detecting anti-Mash2 IgG by ELISA.

**Figure 3 cancers-13-00845-f003:**
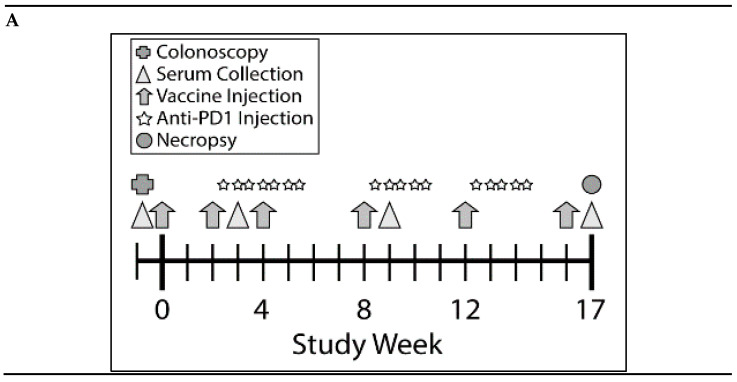
Combination of KISIMA-Mash2 vaccine and anti-PD-1 treatment is safe and promotes T cell immunity in *Apc^+/Min-FCCC^* mice. (**A**) Study design. Animals underwent colonoscopy (cross) prior to enrollment on study. Serum was collected (triangles) at baseline and at three additional intervals throughout the study. Vaccine or vehicle control (arrows) was administered six times, and anti-PD1 or buffer control (stars) was administered in three cycles of three-weekly injections. All animals were necropsied one week after the last vaccine injection. (**B**) Average body weight of animals obtained weekly for the duration of the study. (**C**) T cell immune response assessed by ELISpot on ex vivo splenocytes after overnight stimulation with overlapping peptide pools derived from Mash2 antigen. Bar represents the mean of each group for each peptide pool stimulation. ** *p* < 0.01; *** *p* < 0.001; **** *p* < 0.0001 (2-way ANOVA followed by Tukey’s multiple comparison) (**D**) Humoral response determined in serum at Week 3, Week 9, and Week 17 by detecting anti-KISIMA-Mash2 IgG by ELISA. Line represents the mean for each group. *p* < 0.05 between Vaccinated vs. Control or Anti-PD-1 groups and *p* < 0.001 between Combo vs. Control or Anti-PD-1 groups at Week 17 (Two-way ANOVA followed by Tukey’s multiple comparison).

**Figure 4 cancers-13-00845-f004:**
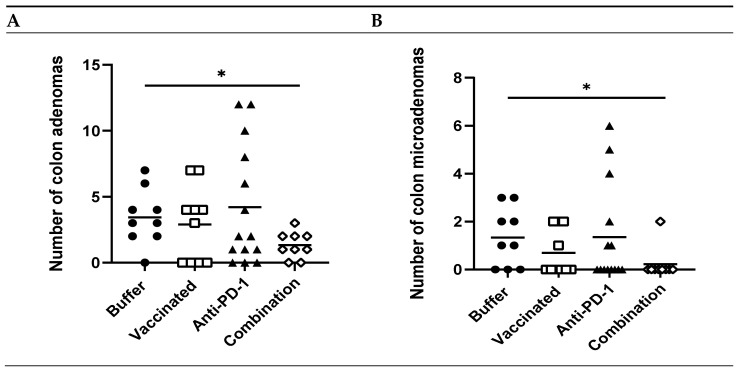
Efficacy of KISIMA-Mash2 vaccine and anti-PD-1 combination treatment on tumor development in *Apc^+/Min-FCCC^* mice. (**A**) Number of colon adenomas per mouse in animals that were tumor-free at treatment initiation (after removal of outliers). (**B**) Number of colon microadenomas per mouse in animals that were tumor-free at treatment initiation after removal of outliers. *, *p* > 0.05. All other statistical comparisons were not significant.

**Figure 5 cancers-13-00845-f005:**
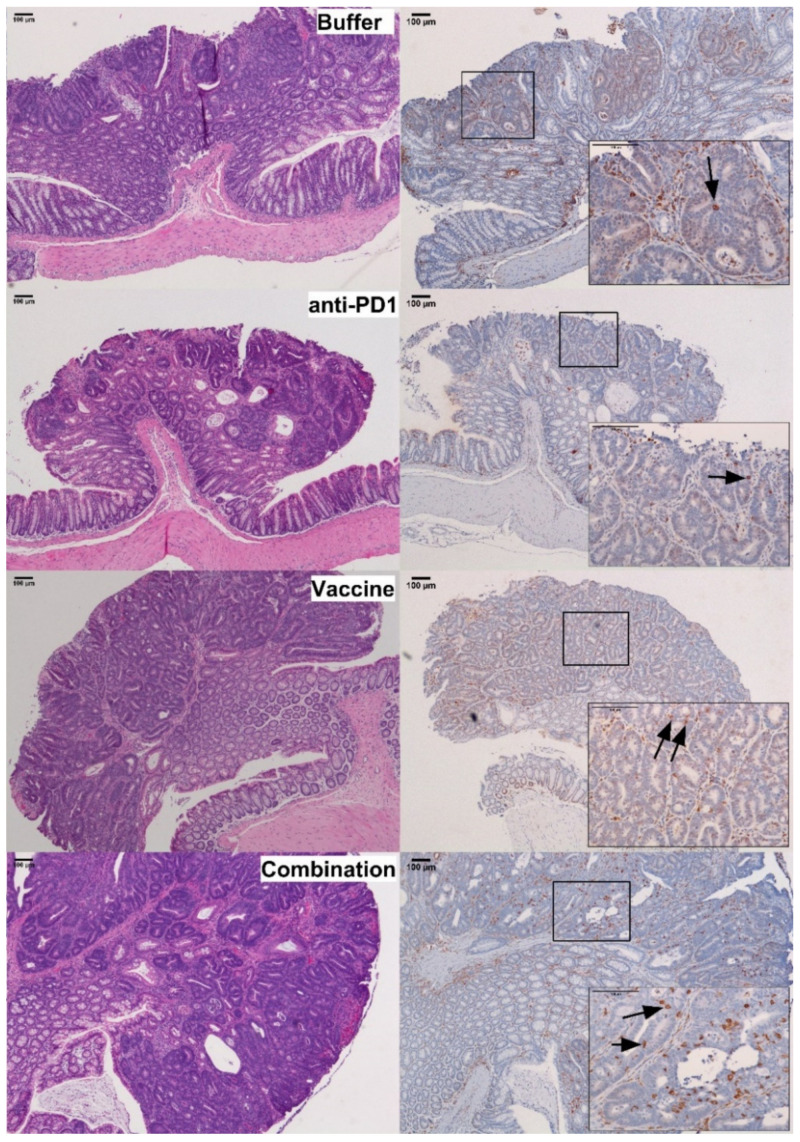
Analysis of CD3+ cells infiltration in *Apc^+/Min-FCCC^* mice. Representative colon adenomas, stained with hematoxylin and eosin (H&E, left panel) and anti-CD3 antibody (right panel), from mice treated with buffer (Buffer), anti-PD1 alone (anti-PD1), KISIMA-Mash2 vaccine alone (Vaccine), or the KISIMA-Mash2 vaccine and anti-PD-1 (Combination). Boxes contain a high power view (200×) of the 40× areas positive for CD3 staining. Representative intraepithelial lymphocytes are highlighted with arrows on the 200× images.

**Figure 6 cancers-13-00845-f006:**
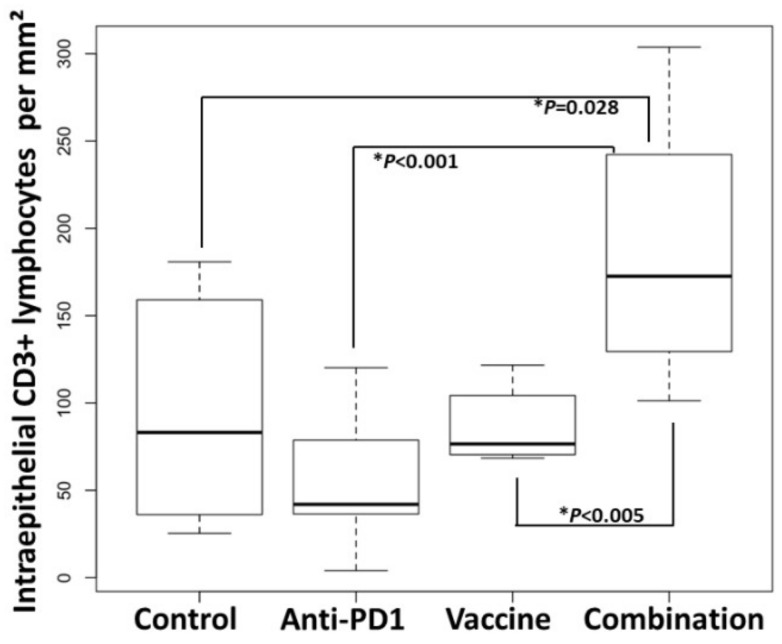
Combination of KISIMA-Mash2 vaccine and anti-PD-1 treatment results in higher T cell infiltration in *Apc^+/Min-FCCC^* mice. Anti-CD3 staining of adenomas from mice treated with buffer (Buffer), anti-PD1 alone ‘anti-PD1), KISIMA-Mash2 vaccine alone (Vaccine), or the KISIMA-Mash2 vaccine and anti-PD-1 combination (Combination). Box plots indicate the number of intraepithelial CD3+ lymphocytes per area (mm^2^) of dysplasia in each animal. Kruskal-Wallis test comparing all 4 groups simultaneously *p* = 0.002. p values from two-sided Wilcoxon tests are shown on the graph.

## Data Availability

Data available on request due to restrictions eg privacy or ethical. The data presented in this study are available on request from the corresponding author.

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
