# Peer review of "Novel Protein-Based Vaccine against Self-Antigen Reduces the Formation of Sporadic Colon Adenomas in Mice"

_cancers, 2021, doi:10.3390/cancers13040845_

Round 1
Reviewer 1 Report
The authors have answered satisfactorily to all my questions
Reviewer 2 Report
The authors have addressed all my questions.
This manuscript is a resubmission of an earlier submission. The following is a list of the peer review reports and author responses from that submission.
Round 1
Reviewer 1 Report
Major comments:
1) The vaccine was administered by sc and the authors checked IFN-gama secreting immune cells in the spleen. The antigen-specific T cells in the tumor should be evaluated, for example, by tetramer staining. This can tell whether the vaccine can elicit tumor-targeting T cells. Especially as the vaccination itself seemed not efficacious in reducing the colon adenomas (Fig.4A-B).
2) The authors indicated that the rationale for the vaccine is for immunoprevention. In the experimental design, PD-1 blockade was evaluated in combination with the vaccine, and only the combo showed efficacy compared to individual treatment.
3) The vaccines had been administered six times during a period of 16 weeks. Is it possible that such frequent immunizations had caused immune exhaustion? The anti-PD1 had also been administered more than 10 times, which is not common in other researches. This antibody clone is a rat IgG2a. Frequent injections may elicit mouse-anti-rat IgG antibodies. Please clarify the rationales behind the individual and combo treatment and the treatment schedule.
Minor comments:
1)Fig.3C, please indicate the Y-axis as IFN-γ secreting cells for clearance.
2)Fig.1 and 2, the serum had been collected multiple times. It would be helpful to show the kinetics of anti-MASH2 titers.
3)Fig.4 It would be helpful to show the statistics for the comparison of other groups.
4)Fig.5 is confusing, please label the images for easier understanding. Also, use boxes to to indicate the zoomed-out regions. eg. Panel D and H.
5) The mouse model seems resistant to anti-PD1 treatment (Fig.4A and B). This should be discussed in the context of previous studies with this model.
6) The Ascl2 is an intracellular protein. The significance of antibody response in the contribution to the vaccine efficacy seems not clear. The authors need to tone down related statement regarding the antibody responses.
Reviewer 2 Report
In this work, Belnoue and collaborators show data on the protection against sporadic colon carcinoma formation by vaccination with a novel protein vaccination platform (KISIMA containing an interesting tumor antigen (Ascl2). They have tested immunogenicity (humoral and cellular) of the vaccine and found that combination of vaccine with anti-PD1 therapy significantly protects from colon adenoma and microadenoma formation in a relevant murine model of colorectal adenoma formation. The results obtained and the new vaccination platform are very promising. The authors published recently the results of this new and promising platform using a multiantigenic domain including several tumor antigens (Belnoue et al JCI Insight. 2019 Jun 6; 4(11). In this manuscript the authors expand their results using the promising tumor antigen Ascl2. Several issues could be addressed before its publication.
- Figure 1 A. Immune response has been measured in control and vaccinated animals by ELISPOT. Splenocytes have been cocultured in the presence of the KISIMA-Mash2 vaccine as an antigen. The authors conclude that vaccination induced an anti Mash2 specific T cell response (lines 243-244). However, the vaccine include other potential antigens in addition to With this experiment, the authors cannot formally conclude that the response is against Ascl2 and not against the cell penetrating peptide or against the TLR2/4 agonist, as they can with the results from figure 3 using Ascl2 peptides.
- Figure 2C. Figure legend indicates that sera from week 22 have used for anti-Mash2 antibody detection. However, this data point is not represented in Figure 3A (Study design).
- Figure 3. Cellular immune response induced by the vaccine is similar to that found by the combination. However, results from figure 4 indicate that only those mice treated with the combination have a protective effect. The authors should discuss in more detail this finding in the discussion section. Are the vaccine primed T cells expressing PD-1? Could it be the reason for this synergistic effect?
- Figure 5. Lines 337-339. The authors conclude that combination therapy (vaccine and anti-PD1) increased T cell infiltration. However, the authors should also show the infiltration level of T cells in the group of mice treated with the monotherapies (vaccine alone or anti-PD1 alone) to conclude that the combination is the responsible of such effect. Does Anti-PD1 alone have a similar effect? Inclusion of these data (from all experimental groups) in Table 1 could help.
- A brief explanation in the methods section of the murine tumor model could be welcomed by the readers
- General comment. Vaccination induced a humoral as well as a cellular immune response. However, the authors does not comment about the relevance of a humoral response against Ascl2, a transcription factor which is not expected to be expressed on the surface of tumor cells and consequently, will not activate an ADCC response or a complement mediated cell cytotoxicity against the tumor. However, the capacity of this vaccine platform to induce a strong humoral response is very interesting and could be exploited for other antigens.
